# Reflections on Sustainable Urban Mobility, Mobility as a Service (MaaS) and Adoption Models

**DOI:** 10.3390/ijerph20010274

**Published:** 2022-12-24

**Authors:** Manuel Rey-Moreno, Rafael Periáñez-Cristóbal, Arturo Calvo-Mora

**Affiliations:** 1Department of Business Administration and Marketing, Faculty of Tourism and Finance, Universidad de Sevilla, 41018 Seville, Spain; 2Department of Business Administration and Marketing, Faculty of Business Economics and Management, Universidad de Sevilla, 41018 Seville, Spain

**Keywords:** sustainable mobility, smart mobility, smart cities, micromobility, mobility as a service (MaaS)

## Abstract

The environmental, social, economic, cultural and demographic changes experienced in a large part of our society are translating into a greater concern for the search of sustainable responses to the concept of mobility. In this context, the main objectives of our study are: (1) to identify the topics that are addressed most frequently in the scientific literature on sustainable mobility, and (2) to analyze the most suitable models of acceptance or rejection of sustainable mobility. The methodologies used in this paper are a literature review and content analysis. This methodology is useful for the objective, systematic and replicable description of scientific literature. The results highlight the multidimensional nature of sustainable mobility and, in turn, its connection with social issues of greater importance, such as the Sustainable Development Goals. Additionally, a conceptual framework is provided on models of acceptance and the use of information systems linked to sustainable mobility.

## 1. Introduction

Forecasts before the pandemic predicted that more than two-thirds of the world’s population will live in cities in 2050, will consume 70% of energy, and will emit a similar percentage of greenhouse gases. These data, together with the necessary global reflections on the causes and effects of COVID-19, will increase social pressure for a greater supply of products and services that improve the sustainability of the planet [1].

Unsurprisingly, the European Commission in its European Sustainable and Smart Mobility Strategy sets the following objectives for the year 2030 [2]: (1) at least 30 million vehicles with zero emissions must circulate on European roads; (2) at least 100 European cities must be climate neutral; (3) to double high-speed rail traffic; (4) to ensure that scheduled collective journeys of less than 500 km are carbon neutral within the EU; and (5) to deploy automated mobility on a large scale.

Closely connected to this issue, the European Environmental Agency revealed, in 2020, the existence of 12 serious environmental problems in the territory of the Union [3]. Among them, there are four that are closely related to this issue: (1) climate change caused by the increase in CO_2_ levels in the atmosphere; (2) acidification resulting from the combustion of fossil fuels and emissions of sulfur dioxide and nitrogen into the atmosphere; (3) forest degradation resulting from atmospheric pollution; and (4) urban stress, such as environmental stress due to poor air quality, excessive noise and traffic congestion. With the increase in residents and visitors in cities, societies must address urgent challenges such as mobility, both because of its impact on the quality of citizen life and because of the externalities it generates (pollution, noise, congestion, etc.) [4,5,6]. We cannot forget that transport within the European Union is a key economic sector that generates a gross added value of EUR 599,000 million for transport and storage services, representing 5% of the total gross added value of the European Union in 2018.

This sector represents 19.5% of total greenhouse gas emissions and is the only one that has not experienced a decrease in these emissions between 1990 and 2018. In 2019, transport represented 30.9% of final consumption of energy [7]. Although there is increasing awareness of this problem [8], there is still a large gap between consumer environmental attitudes and their consumption behaviors [9]. Although public and private organizations insist on the need to adopt the 7 Rs model (Reduce, Reuse, Recycle, Redesign, Renovate, Repair and Recover), there is still a long way to go before this circular economy mechanism is incorporated in the daily life of people, companies and cities [10].

In this context, we identify the following research questions (RQs):RQ1: What topics are addressed most frequently in the scientific literature on sustainable mobility in order to frame this phenomenon and analyze its different consequences?RQ2: Considering the importance of adoption models in technological products, what theoretical framework could be most useful for conducting future research into this issue?

The answers to the research questions make it possible to fill gaps identified in the literature. On the one hand, the issues usually addressed in studies on sustainable mobility are related to the difficulty of understanding a complex multidimensional concept. On the other hand, for empirical purposes, sustainable mobility will only be achieved if people develop behaviors that lead them to internalize and adopt the technological products that support the achievement of the aims pursued. In this sense, the analysis of the adoption models of these products should provide a basic theoretical framework for future research within the scientific community.

Trying to answer the previous research questions, our work begins by describing the methodology used. After the methodological considerations, the following section is dedicated to the analysis of the results obtained. At this point we discuss the relationships between mobility and the concept of the smart city, as well as the issue of micro mobility and its connection with the use of private vehicles. Our work also dedicates a paragraph to discussing the linkages between sustainable development, mobility as a service (MaaS) and marketing. Before elaborating on the conclusions, the research highlights the importance of studying exploratory adoption models and their influence on sustainable mobility. Finally, the main conclusions and limitations of the study are presented.

## 2. Materials and Methods

The review of the literature and the content analysis as the methodology of this study, has its origin in the research questions proposed, and follows the guidelines developed by studies with similar objectives, such as those of [11,12]. The content analysis is based on reading as an information-gathering instrument. This reading must be carried out following a scientific method, that is, it must be systematic, objective, replicable and valid [13]. According to [14], a content analysis is a research technique for the objective and systematic description of the content of communications, in order to interpret them. More specifically, the existing scientific literature on “sustainable mobility”, “smart mobility”, “smart cities”, “micro mobility” and “mobility as a service” (MaaS) was identified, analyzed and reviewed. The databases used were the two most important that currently exist, due to their wide coverage and quality of content: Web of Science and Scopus.

The Web of Science (WOS) database is an online platform that contains databases of bibliographic information and information analysis resources that allow for the evaluation and analysis of research performance. Its purpose is not to provide the full text of the documents, but rather to provide analysis tools that help to assess their scientific quality. It helps to access to different databases through a single query interface, being able to access a single database or several simultaneously. Its content is multidisciplinary and provides information of a high academic and scientific level. Specifically, the following database collections were used: Science Citation Index Expanded (SCI-EXPANDED), 1900-present, and Social Sciences Citation Index (SSCI), from 1956 to the present.

The Scopus database is the largest abstract database seen in the world to date, with 20,500 publications from more than 5000 international publishers, and with access to more than 28 million abstracts (since 1966) and 5 retrospective years of references (reaching 10 years in 2005). It represents approximately 80% of international peer-reviewed publications, ensuring up-to-date content as a result of its weekly updates. Therefore, it acts as a simple and unique access point for users that is “as easy to use as Google”, according to the company, offering the fastest access to the full text of research articles. It enables the best navigation through the available scientific literature thanks to its new search and functionality. Specifically, the collections of (1) Physical Sciences were used, which contains more than 7200 titles, including those related to engineering and environmental sciences, and (2) Social Sciences & Humanities, which contains more than 5300 titles related to business, accounting, management, economics, finance or decisions science.

The papers used in the content analysis were selected according to the following criteria [15]:Temporary scope. Papers published between 2016 and 2022 were searched for and selected. Specifically, in 2016 the first MaaS application appeared.Quality of research. Only scientific articles published in journals available in the two main databases mentioned above were selected.Knowledge area. Sustainable mobility is a multidisciplinary field of study. For example, sustainability, marketing, psychology, logistics and transportation, geography, energy, technology, among others. This leads us to use databases related to different areas of knowledge.Language of publication. Papers published in English are analyzed.Keywords. Taking the research questions as a reference, the following keywords were used: “sustainable mobility”, “smart mobility”, “smart cities”, “micro mobility”, “mobility as a service”, “MaaS”, and “technology acceptance models”.Search strategy in the databases. The combination of the keywords in the databases was carried out in the following fields: subject (for the WOS) and titled, abstract and keywords (for Scopus). In addition, it was limited by publication date (2016–2022), and English language.

Regarding the results and due to the significant number of papers that met the search requirements (324), we firstly proceeded to a detailed reading of abstracts. Secondly, for the works that seemed to meet the objectives set out in the research (146), the complete paper was analyzed. The result was the selection of 78 works to carry out the content analysis. Finally, the journals that publish the most on the subject matter are: Sustainability, Technological Forecasting & Social Change, and Transportation Research Part A: Policy and Practice.

## 3. Results

### 3.1. Smart Cities and Mobility

There are several trends that are currently observed related to sustainability in urban areas. Undoubtedly, one of the most relevant are the so-called smart cities (Smart Cities), an issue that is attracting increasing interest from researchers, especially since the second half of the last decade [16].

Smart Cities are ecosystems that stand out for their holistic vision [17], where multiple agents interact to implement innovative solutions [18] and in which Information and Communication Technologies (ICTs) are useful instruments in achieving higher levels of sustainability [19,20]. Within the classic model of Smart Cities, mobility has always been highly prioritized. It is typical to consider Smart Mobility among the pillars of the Smart City, either as a category [21] or as an infrastructure [22].

In this context, Smart Mobility is one of the pillars of a Smart City and as a concept aims at improving the quality of life of citizens [4]. It includes intelligent urban transport networks [3] and focuses on the use of systems that improve urban traffic and sustainability in this area [23]. The so-called Intelligent Transport Systems consist of a set of technologies and applications aimed at improving safety and mobility in traffic, as well as increasing labor productivity and reducing the negative impacts of traffic [24]. The definition proposed by [25] states that Smart Mobility is the result of a planning process that makes use of technological supports in the simulation, use and monitoring phases of individual and shared transport systems to guarantee safety standards, functionality and sustainability. It is focused, therefore, on the use of integrated ICTs, sustainable transport and logistics systems, to support better urban traffic and mobility [5].

In this context, as noted in [26], service quality in smart cities represents a complex issue that requires a new set of skills and measurement and planning methods. Those related to the use of graphic traffic management mechanisms should be included [27]. 

The following Table 1 schematically represents the attributes of Smart Mobility:

### 3.2. Private Vehicles versus Shared Micromobility

One of the issues linked directly to Smart Mobility is the use of private vehicles in urban areas as an unsustainable form of transport. Both human-driven vehicles and self-driving or autonomous driven vehicles are included in this section [28]. In this regard, the change in favor of ecological systems is increasingly observable in society, which is linked to current concepts of urban micro mobility [29]. According to [1], shared micro mobility is an integrated multimodal mobility solution that can play a significant role in the development what is termed as Vision Zero: zero pollution, zero emissions and zero congestion. Shared micro mobility is, therefore, an innovative transportation strategy that allows users to access modes of transportation in the short term based on the fulfillment of an on-demand need, and that includes various types of services and modes of transportation [30].

Although the COVID-19 pandemic has had beneficial effects in several aspects related to the reduction in mobility due to the rise of teleworking [31] or the greater public awareness of respect for the environment [32], in other ways it has meant a clear setback. This is the case for certain elements linked to Smart Mobility. In the years prior to the pandemic, there was a gradual and growing tendency in the world population to use instruments such as shared mobility systems to the detriment, for example, of the individual car [33]. However, the pandemic has meant an evident setback in this field due to fears of contagion associated with the idea of carpooling or the use of public or collective transport [34]. It can be assumed that this attitude may be temporary and not permanent depending on the pandemic situation disappearing, leading to a return to past behaviors and trends.

Therefore, the citizen debate and the doubts of the population regarding the dilemma of the use of a private vehicle or public transport (and the corresponding fuel savings [8]), or the adoption of other consumption models that combine means of transport defined through technology and community use [35], remain. In these models, the consumer acquires time for use and consumption, with the possibility of accessing objects or services that otherwise are not affordable or that they decide against, due to limitations related to space or the environment [36,37].

Shared mobility implies the shared use of a vehicle which is accessed in the short term when it is necessary, and complements traditional public transport services [5,38,39]. This type of mobility is based on the collaborative economy and the provision of on-demand service systems [4,37]. The collaborative economy emerges as a disruptive approach to doing business [39]. The use of platforms is common and is linked to sustainable ways of acting and social change [40,41], being effective in motivating people to develop behaviors and share information [42]. They usually facilitate disintermediation, which makes them very attractive in the eyes of consumers [43].

Within sustainable and ecological urban mobility solutions based on the sharing economy, the use of electric vehicles has attracted interest worldwide [44], due to users receiving the benefits of a private vehicle without the costs and charges of owning one. Although its use will lead to the need to generate new urban infrastructures in the future [45], there are multiple benefits associated with this alternative, both individual and collective [46].

### 3.3. Sustainable Development and Mobility

The research in this field would necessarily be linked to the development of ICTs, as well as the Sustainable and Shared Mobility Strategies, which is one of the great challenges in the context of implementation of the 2030 Agenda, as well as the Sustainable Development Goals (SDGs). Achieving sustainable mobility is, probably, one of the most difficult tasks among the myriad challenges associates with sustainable development [47].

The 2030 Agenda for Sustainable Development marks the roadmap for advancing in this model for the next decade. It was approved by the United Nations Assembly in 2015 on September 25, in the resolution “transforming our world: the 2030 Agenda for sustainable development”. Their objective was to stimulate action for five elements of maximum importance, which they called the five Ps (people, planet, prosperity, peace and partnership).

Based on the experience with the Millennium Development Goals (MDGs), one of the main results of the Rio + 20 conference held in 2012 was the agreement to develop a set of SDGs on which the 2030 Development Agenda is based. Although it does not represent an independent SDG, there are multiple aspect related to Sustainable Mobility included in some of the 17 SDGs. To delve deeper into this issue, it could be indicated that it is included, at least, in three of these SDGs. Firstly, in SDG7 (“ensure access to affordable, safe, sustainable and modern energy for all”), secondly in SDG9 (“Build resilient infrastructures, promote inclusive and sustainable industrialization and foster innovation”) and, finally, SDG 11 (“Make cities and settlements inclusive, safe, resilient and sustainable”). In all three of them, especially in SDG7, the transformation of mobility and transport into more sustainable models is essential.

The decarbonization of the sector involves prioritizing investment in public transport, promoting non-motorized means and electrifying mobility in cities since, among other effects related to sustainability, electric mobility can physically modify the urban fabric of cities and restore public spaces, in addition to creating new ones [48].

In the specific case of Spain and beyond the 2030 Agenda, the Spanish Strategy for Sustainable Mobility (EEMS) emerges as a national reference framework that integrates the principles and coordination tools to guide and give coherence to sectoral policies that facilitate a sustainable low-carbon mobility. The objectives and guidelines of the EEMS are specified in 48 measures structured in five areas, including the planning of transport and its infrastructures [49]. Among the measures contemplated, special attention is paid to the promotion of alternative mobility to private vehicles, the use of public transport, the use of more sustainable fuels [50,51], while considering urban planning and mobility infrastructures as key elements.

### 3.4. Sustainable Mobility and MaaS 

Previously, we noted that one of the fundamental pillars of the smart city is Smart Mobility, which focuses on the use of transport systems to support urban traffic and sustainable mobility [5]. As an evolution of this dimension, MaaS represents an attractive alternative to the possession and use of a private vehicle in which the mobility intermediary plays a determining role [52]. In fact, MaaS is often referred to a system in which different mobile operators provide users with different services. In MaaS, ICTs are the means to offer an integrated service that mixes private and public transport, while using other means of transport which provide the advantage of sustainability (taxis, bicycles, rental cars, shared trips, and so on). Within MaaS, all activities related to itinerary planning and the provision of reservation services, ticket purchases or payment are also included. In this way, MaaS guarantees that the following two major problems faced by public transport users are eliminated: exchange gaps and insufficient services (car parks, for example) [53]. MaaS combines different modes of transport to offer consumers the possibility of following a route in a flexible, personalized way, on demand and without interruptions, through a single interface [54]. It has been pointed out as one of the strategies to apply to achieve sustainable tourism [55,56]. MaaS is therefore the combination of various shared mobility trends (car sharing, motorcycle sharing, bike sharing, etc.) together with public transport (taxis, buses, commuter trains, etc.) [57] as an alternative to the private vehicle. In these services, the use of shared electric vehicles is becoming increasingly significant, which will allow for more sustainable urban mobility. 

The MaaS concept includes three key components and has five possible levels of integration [58]. Regarding the components, the following are indicated: (1) Provision of a defined service focused on the transport needs of the customer, user or passenger; (2) provision of mobility, not transport; (3) integration of transport services, information, payment and a ticketing system.

Regarding the levels of integration, five are proposed: (1) Level 0: no integration (the lowest level refers to a situation where separate services are provided for all types of transport without any kind of integration); (2) Level 1: integration of information (the added value of level 1 is the support in making decisions for the best trip, the optimal route, the ease in choosing the time of day, which route to take and which transport to use); (3) Level 2: integration of reservation and payment (helps transportation service providers to reach more customers); (4) Level 3: integration of transport services in packages (the goal is to fully cover the customer’s transportation needs and provide a large number of services that is impossible for lower levels); (5) Level 4: integration of public objectives—Smart City (the use of the private vehicle decreases drastically and transport services are much more accessible than in a non-Smart City.

The implementation of this service between users is usually through applications and digital platforms that offer a comprehensive shared public transport solution that is committed to multimodality. In Europe and North America, some cities have chosen to include these apps in their transport services, although as of 2021 it could be seen that their use worldwide was still quite limited [59].

MaaS, therefore, combines public transport mobility services (taxis, car rentals, car sharing, bicycles, motorcycles and scooters, for example) in a single platform that can be accessed from a smartphone. A MaaS platform not only plans the trip, but also allows users to purchase tickets from a wide variety of service providers. MaaS raises the possibility of reducing urban congestion and cleaning the atmosphere while helping users and minimizing the impact on the environment of the use of vehicles. With the increasing adoption of electric mobility and the expanding development of this solution, the possibility of following a multimodal route, without interruptions and in a more ecological way is realized [50]. For the implementation of these systems to be successful in large cities, there are several key elements to consider. Among them, the following three stand out: (1) Infrastructure that provides increasing, effective and efficient circulation systems. (2) Availability of data that provides knowledge on the habits of citizens, the busiest urban areas and zones, the hours of greatest traffic, etc., in order to be able to plan both in the short and long term. (3) Direct and indirect incentives that help catalyze change to new systems, which requires support from institutions and public urban planning and mobility policies that favor the use of these services.

The services to these platforms are accessible through the web and mobile apps. The first application of this service was made in 2016 by MaaS Global in Helsinki. It was called Whim and included public transport services, taxis, car rentals, shared cars and bikes. In addition, the MaaS options also include (1) Meep (https://meep.app/es/) (accessed on 17 November 2022), which personalizes the routes through an algorithm that identifies tastes, based on the preferred options indicated by the user (cheap, fast or ecological), and on real-time information from the platform operators; (2) Moovit (https://moovitapp.com/) (accessed on 17 November 2022), which integrates shared cars, motorcycles and bicycles, electric scooters and taxis. Users receive information to plan their routes, reserve and buy the different modes of transport. Among its services, Moovit has incorporated an augmented reality service with the Way Finder function.

Regarding the adoption and implementation of MaaS in the future, uncertainty is high. In this regard, the research carried out in [60] is interesting, as it provides a compilation of the latest research carried out in which different scenarios regarding the future of Maas are presented.

### 3.5. Marketing and Mobility

The environmental, social, economic, cultural and demographic changes experienced in a large part of society are translating into a greater concern for the search for sustainable responses within the concept of development that has been considered up to now. The growing impact of these trends on marketing thought and practice is being reflected in the way consumers interact with companies, thereby giving rise to what is referred to as transformative marketing [61,62]. This approach is aligned with research in the field of consumer behavior and problems of great social impact [63,64]. One of the forces that supports this approach is the pressure faced by public administrations and private organizations regarding the management of natural resources, as well as global social and health crises [65].

The marketing 3.0 concept was conceived of in Asia in 2005 by Hermawan Kartajaya. After two years of improvement, it was presented by Phillip Kotler and Hermawan Kartajaya at the 40th anniversary of the Association of Southeast Asian Nations in Jakarta. Finally, Iwan Setiawan collaborated with Kotler in its development, and shaped it in the framework of a new digital scenario [66]. Today’s society seeks spiritual resources over material satisfaction; therefore, consumers are no longer looking only for products and services that meet their needs but are also concerned with identifying the business models that could be behind the organizations that manufacture the goods used for it. A values-centric business model underpins marketing 3.0 [66].

Marketing 4.0 is a natural evolution of marketing 3.0 that has been extended with the following four powerful elements: (1) brand identity, (2) brand image, (3) brand integrity and (4) brand interaction [67]. In this new scenario, the final objective of marketing is to guide consumers from the initial phases of attention and discovery of a good, to later stages beyond the behavioral loyalty (repeat purchase), related to recommendations to other consumers (emotional fidelity). The establishment of emotional ties with the customer is the basis of this defense of the company, and the identification with spiritual, social, human, and environmental values is a good sign of commitment [68]. Following these premises, the authors present a society with horizontal, inclusive and social behaviors, which means that those organizations with responsible and sustainable business models hold great potential to achieve solid competitive advantages [9,69]. Integrating the concepts of marketing 4.0 and potential users of sustainable mobility, the findings made in other contexts by [70,71] could be extended in the sense of affirming the idea that identity and brand image are significant factors in determining satisfaction of the client and the intention of use.

With the general acceptance of the key principles of marketing 3.0, after more than a century, marketing has managed to overcome its product orientation to develop on the concept of human centricity. Marketing 3.0 may be considered the last stage of traditional marketing. Nowadays, marketing approaches have moved from an anthropocentric paradigm centered on the human being, i.e., from an intellectual (marketing 1.0), emotional (marketing 2.0) and spiritual (marketing 3.0) point of view, to a biocentric one, focused on society and nature with the support of digital technology (marketing 4.0 and 5.0).

This approach seeks to identify factors that influence consumers to behave responsibly towards the environment, substituting the consumption of products with a greater impact on the planet [69,72,73,74,75,76,77]. The work in [78] defines the values of this consumption as the tendency to explore the value of protecting the environment through purchase and consumption behaviors. Therefore, consumers with values of green or ecological consumption are, generally, more oriented towards protecting resources and purchasing responsibly [79].

### 3.6. Sustainable Mobility and Explanatory Models of Adoption 

The present work could be useful in the future, for example, in the realization of new empirical investigations in which the general aim is to identify the profile of the current and potential segments of users of MaaS services in a certain urban spatial context, as well as the factors that determine and could anticipate its adoption. Although interesting theoretical studies have been carried out based on the scientific literature of the subject to identify the expected and desired effects of the use of MaaS as well as the presumed barriers associated with its use (both on the supply side and on the demand side) [80], further empirical research on the matter is still lacking.

Research related to the acceptance of new information systems basically focuses on their initial adoption, with the understanding that their effective use is a consequence of the intention to use [81] and that this is shaped by evaluations of the individual in the pre-adoption phase [82]. While the criteria prior to adoption are generated by indirect external stimuli, the subsequent criteria are based on personal experience [83]. The explanation of adoption and use is usually based on the paradigm of planned behavior, which accepts that behavior is a conscious process that, in addition to variables such as price [84], is influenced by beliefs, attitudes and behavior intentions [85]. The intentions show attitudes towards the behavior, defined as the favorability towards the consequences of an act, in addition to the relevance that its effects have. From this approach, attitudes are related to behavior through the effect of their intentions.

As for the choice of a conceptual framework for such research, there are different options. The research carried out in [86] classifies these possible conceptual frameworks into three blocks. The first revolves around three conceptual axes (sociodemographic characteristics, attitude preferences, and mobility and travel patterns) and their extensions (built environment, psychographics, geography, climate, personal traits, etc.). The second one is linked to endogenous and exogenous factors and, finally, the family of Unified Theories of Acceptance and Use of Technology (UTAUT).

In the context of UTAUT, the application of dual adoption models which contemplate the so-called facilitating and inhibiting factors of adoption, has gained relevance in recent years [87]. The dual factor theory posits that consumers are influenced by two distinct blocks of impulses, which favor or frustrate the adoption of innovations [87]. Thus, while facilitators both favor and hinder adoption, inhibitors disfavor it when they are present, but do not necessarily activate it if they are absent. Inhibitors and facilitators must be considered as different and independent constructs that can coexist, while presenting different antecedents and consequences [88].

There are multiple proposals on realities that stimulate or block sustainable transitions in urban spaces [89], as well as on the facilitating factors of personal adoption in information systems [90]. They all share a rational vision and highlight the role of beliefs as active determinants of the users’ attitude, these being in turn key determinants of use. UTAUT1 [91] can be considered as a successful compendium of previous theories that maintains broad support today. It establishes that the intention to use is determined by the expectation of results, expectation of effort, social influence and facilitating conditions. As a continuation of this model, [92] proposed a new model several years later that extended the previous one and which they called UTAUT2. It added hedonic motivations, perceived value and, finally, to the previous variables included in UTAUT1, the habit. In the same UTAUT context, studies have subsequently continued to analyze the reasons for acceptance and the profiles of users of MaaS applications or of specific elements of sustainable mobility [93,94,95,96]. They show that the psychological factors of UTAUT1 and 2, as well as attitude, social influence and sociodemographic and transportation-related characteristics, among others, act as powerful predictors of behavioral intention.

Habit is a construct widely considered in the literature as an influential element in the behavior of individuals in the face of innovations. It highlights the importance of the automatic use of a system beyond planned, conscious or reasoned behavior. These works lead to a second behavioral paradigm: the so-called object-oriented automaticity. The study in [97] describes the interrelation between both paradigms suggesting that action and control follow two basic processes, namely the repetition of past actions, as well as the conscious effect of attitude and intentions. This proposal means that both habit and intentions (the latter guided by attitude) jointly predict behavior, and may sometimes coincide in terms of the effects or, on the contrary, they may come into clear conflict. As noted above, there are times when habits collide with attitudes, thus giving rise to so-called counter-intentional habits. The study in [98] posits two combinations of variables that reflect positive or negative intention and behavior that may or may not be outcome-based. When habit and attitudes coincide, they lead to the same result; however, when they contrast one another, the result of the behavior depends on the strength of the habit or the intention [99]. There are several studies that confirm that prior mobility essentially determines what requirements a MaaS system must meet to be considered a suitable alternative [100]. It is convenient, therefore, to consider that it is quite complex for those individuals who develop strong habits to change their behavior, no matter how rational the alternative that is presented to them and the utility they receive is.

One of the unconscious mechanisms influencing the consumer’s decision that has great relevance is resistance to innovation [101]. It is defined as the tendency to avoid making changes [102]. It can also be considered as the rejection that individuals show in the face of innovation, because it can lead to changes in their current state of satisfaction, or because it can generate a conflict in their belief structure [103]. In the field of information systems, resistance is defined as an adverse reaction or strong opposition to change to a new alternative [82]. As the adoption of a new system results in the replacement of an old one that has been used, resistance is explicit or manifested as the reluctance to change in terms of the technology used [101].

There is a high degree of agreement in the literature on facilitators. The same is not the case with inhibitors. A theory that explains how the use of a system could influence behavioral intentions towards a new one is known as the Status Quo Bias (SQB). This theory has been extensively studied in the literature [104,105,106,107]. It states that when an individual is faced with various alternatives, they usually choose the one among all those offered that is linked to their current status quo, even in the presence of other superior options in terms of value or utility. It is a tendency of a cognitive or rational nature, dependent on the context, in favor of the present situation [97], and that attempts to explain the influence of maintaining behavior through inhibiting perceptions on the use of the new system [87]. 

Both resistance and SQB manifest as inertia [101,103]. Inertia describes a behavioral tendency to rely on what was previously chosen and what the current state represents. Inertia-driven individuals avoid seeking variety and innovation. Inertia is an unconscious, expediency-driven emotion that suggests that repeated use is carried out passively, without thinking about it or keeping in mind the negative perceptions associated with it. In other words, inertia reflects a rigid continuation of the status quo [101]. In the field of public services, the existence of a segment of citizens who would find it very difficult to change their behavior due to inertia [108] is verified. 

Therefore, there are numerous factors considered inhibitors that discourage the adoption of new technological products. Due to this, and although there are multiple innovations that fail due to the non-acceptance of users, it cannot be said that there is a modeling method most widely accepted in academia as a central reference of these inhibiting factors. In other words, it is not the same as in the case of the facilitators where, as indicated, there was a certain consensus around the UTAUT1 model. On the other hand, it is worth pointing out a long list of constructs in which this influence has been demonstrated. Among them exist habit, resistance and inertia (already indicated). Some authors have explored additional factors. Thus, [82] focuses on resistance and antecedents such as the benefit of change, costs of change, perceived value, self-efficacy, organizational support, and peer opinion. Reference [87] proposes sunk costs, loss aversion, inertia, perceived value, transition costs and uncertainty. Some others focus on inertia, habit, sunk costs, and transition costs [101]. The latter approach is very similar to that of [109], in which habit, costs of change and satisfaction. We finish with an exploration of the proposal of [110], which indicates the endowment effect (giving more value to one’s own assets) related to risk aversion, as well as the anchoring effect linked to sunk costs (assessment of the new system based on recent experiences).

Taking as a reference the SQB Theory and its link with the resistance and inertia constructs, the works of [92,101,109,110] identify numerous factors that act as inhibitors to change. The UTAUT2 model [92] includes, in addition to the UTAUT1 facilitators, some recognized inhibitors. Both the socioeconomic characteristics of the individual and the attributes related to the trip can be explanatory factors for the adoption of shared electric vehicle services. Age and education may be variables that explain its use [4]. This is confirmed in the exhaustive research carried out in [100], where it is stated that the citizen strata most likely to use MaaS systems are young, progressive and well-educated people. In the same way, but in relation to the territorial context, the research in [111] shows that the use of micromobility is more frequent in places where the level of purchasing power is greater, the distances to be covered are shorter and, consequently, the car is used less. It happens, however, that these individual factors on multiple occasions only explain a small proportion of the environmentally responsible behavior of consumers, and other variables—such as, for example, values—may be more influential in this behavior [112]. Values reach beyond specific moods of the individual. They must be defined as stable beliefs that favor the execution of certain actions and that lead to desired final states. People’s values are abstract and deeply rooted in motives that explain their attitudes, opinions, and behaviors [112,113].

## 4. Conclusions

Our work has made it possible to frame the concept of sustainable mobility in the context of the subjects that are most frequently associated with this construct.

Issues such as smart mobility, smart cities, micromobility, shared mobility or MaaS appear when reviewing the scientific literature on the subject. This is a consequence of the multidimensional nature of the phenomenon studied and, in turn, its connection with social issues of a broader spectrum or of greater public importance, such as sustainable development or the SDGs. Likewise, the close connection between the tendency to use elements which favor sustainable mobility and marketing 3.0 and 4.0, connected with ecological or green marketing, is highlighted.

Regarding the models of acceptance and use of information systems linked to sustainable mobility, our work provides an open conceptual framework that, after proposing different options extracted from the scientific literature on the subject, is positioned in favor of dual models of adoption. Research on factors which stimulate or slow down the citizen adoption of different elements of sustainable urban mobility could have an adequate reference in this framework, which would help to frame new findings on the subject.

The study is a theoretical and exploratory one and, although it can be observed as a limitation, we must not forget that our aim is to reflect on sustainable mobility and its adoption. Thus, future research should propose bibliometric analyses and empirical studies to continue advancing in the knowledge of this emergent field of study.

## Figures and Tables

**Table 1 ijerph-20-00274-t001:** Attributes and meanings of Smart Mobility.

Attributes	Meaning
Flexibility	Allows users to choose between multiple modes of transport that suit their needs through a navigation system
Efficiency	Provides efficient mobility options with minimal disruption, low cost, and minimal travel time
Integration	Guarantees end to end route plans, regardless of modes of transport
Sustainability	Promotes cleaner, more sustainable operations with minimal greenhouse gas emissions
Safety and Protection	Helps to improve road safety
Social Benefits	Offers citizens equal opportunities to use public transport
Automation	Facilitates the automation of all processes
Connectivity	Generates a network of connected entities
Accessibility	Is affordable to everyone
User Experience	Ensures a better user experience

## Data Availability

Not applicable.

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
