# Peer review of "Reflections on Sustainable Urban Mobility, Mobility as a Service (MaaS) and Adoption Models"

_ijerph, 2022, doi:10.3390/ijerph20010274_

Round 1
Reviewer 1 Report
In the present manuscript, the authors review several aspects related to the sustainable urban mobility and analyze the most suitable models. They discuss different topics like: smart cities and mobility, private versus shared micromobility, models of adoption.
The proposed topic of research presented in this manuscript is interesting, and the presentation quality is generally good. I think that an index should be added at the beginning of the paper, to improve readability.
It would be attractive to include the techniques presented in this paper with some recent results about traffic models ("Phys. Rev. E 64, 056132 (2001)", "SciPost Phys. Core 5, 022 (2022)", Complex Dynamics of Traffic Management [ISBN: 978-1-4939-8763-4] (2019)). It would attract the attention of experts working in similar areas and make the paper results more complete.
I do recommend the paper for publication after the suggested changes.
Author Response
Respond to reviewer
Manuscript ID: ijerph-2095184. " Reflections on Sustainable Urban Mobility, Mobility as a Service (MaaS) and Adoption Models"
The authors are thankful to the Editor and the Reviewer for their insightful comments. The Editor and Reviewer have detected the manuscript's weaknesses and made appropriate comments for correcting them. Likewise, such comments and suggestions significantly improve the clarity of their manuscript. The authors hope to satisfy the standards of IJERPH in its corrected version.
Reviewer #1
#1.1. The proposed topic of research presented in this manuscript is interesting, and the presentation quality is generally good.
R: We are very grateful to the reviewer for this assessment.
#1.2. I think that an index should be added at the beginning of the paper, to improve readability.
R: We thank the reviewer for the appreciation. We have included and index (at the beginning) in the updated version of the paper.
#1.3. It would be attractive to include the techniques presented in this paper with some recent results about traffic models ("Phys. Rev. E 64, 056132 (2001)", "SciPost Phys. Core 5, 022 (2022)", Complex Dynamics of Traffic Management [ISBN: 978-1-4939-8763-4] (2019)). It would attract the attention of experts working in similar areas and make the paper results more complete.
R: The authors thank the reviewer for this comment. Following the reviewer´s suggestion and trying to call the attention of experts in related field, the three suggested references have been cited with the inclusion of the corresponding comments. They can be found in sections 3.1 and 3.2 of the updated version of the article.
Reviewer 2 Report
I would like to congratulate the authors for their work for their great work. I do however have some comments that I think will need to be addressed to help improve the manuscript.
First, the authors need to more clearly explain the research gap that can be addressed by answering the two proposed research questions. As it stands, it is difficult to understand how answering the proposed research questions will contribute to existing literature.
Second, the authors did not include a summary of the layout of the paper at the end of the introduction section. A brief description explaining each section of the paper should be included.
Third, although the authors did provide an explanation of the content analysis used, they did not provide an explanation to the literature review methodology adopted. The literature review also needs to be done in a scientific and systematic way. What systematic approach was used to conduct the literature review? For example, the State-of-the-Art Matrix analysis is one systematic literature approach that can be used.
Fourth, the authors need to provide more detailed information regarding the literature reviewed. For example, How were relevant publications found and identified? What were the search criteria used? Any specific keywords used? How many publications were considered? From how many journals or proceedings? What is the time period of these publications? As it stands, most of the methodology section focuses only on explaining the Web of Science and Scopus databases not the methodology used in identifying relevant literature.
Fifth, in Table 1, how were these attributes identified? Also, the title of table 1 is in Spanish. It needs to be translated to English to be consistent with the language used in the rest of the manuscript.
Sixth, the authors need to further explain the concept of Sustainability as a Service (MaaS) for reader who are not familiar with the concept.
Finally, I think the paper will benefit greatly from a comprehensive review of grammar and spelling mistakes.
Thank you and looking forward to seeing the revised manuscript.
Author Response
Respond to reviewer
Manuscript ID: ijerph-2095184. " Reflections on Sustainable Urban Mobility, Mobility as a Service (MaaS) and Adoption Models"
The authors are thankful to the Editor and the Reviewer for their insightful comments. The Editor and Reviewer have detected the manuscript's weaknesses and made appropriate comments for correcting them. Likewise, such comments and suggestions significantly improve the clarity of their manuscript. The authors hope to satisfy the standards of IJERPH in its corrected version.
Reviewer #2
I would like to congratulate the authors for their work for their great work. I do however have some comments that I think will need to be addressed to help improve the manuscript.
R: We are very grateful to the reviewer for the comments.
#2.1. First, "The authors need to more clearly explain the research gap that can be addressed by answering the two proposed research questions. As it stands, it is difficult to understand how answering the proposed research questions will contribute to existing literature".
R: We thank the reviewer for this appreciation. We have proceeded to incorporate an explanatory text, in response to the reviewer´s suggestion, after the two research questions. We think that the new text covers the aim stated by the reviewer.
#2.2. Second, "The authors did not include a summary of the layout of the paper at the end of the introduction section. A brief description explaining each section of the paper should be included".
R: We thank the reviewer for this suggestion. The new version of the paper includes a brief description of the sections in which the article has been structured, at the end of the introduction section.
#2.3. Third, "Although the authors did provide an explanation of the content analysis used, they did not provide an explanation to the literature review methodology adopted. The literature review also needs to be done in a scientific and systematic way. What systematic approach was used to conduct the literature review? For example, the State-of-the-Art Matrix analysis is one systematic literature approach that can be used".
R: The authors thank the reviewer for the comment. The methodology followed for the bibliographic review is explained in section 2 (Materials and Methods).
#2.4. Fourth, "The authors need to provide more detailed information regarding the literature reviewed. For example, How were relevant publications found and identified? What were the search criteria used? Any specific keywords used? How many publications were considered? From how many journals or proceedings? What is the time period of these publications? As it stands, most of the methodology section focuses only on explaining the Web of Science and Scopus databases not the methodology used in identifying relevant literature".
R: The authors thank the reviewer for the suggestions. In section 2 (Materials and Methods), an attempt has been made to provide answers to the questions raised by the reviewer.
#2.5. Fifth, "In Table 1, how were these attributes identified? Also, the title of table 1 is in Spanish. It needs to be translated to English to be consistent with the language used in the rest of the manuscript".
R: We thank the reviewer for this comment. This error has been corrected in the updated version of the paper.
#2.6. Sixth, "The authors need to further explain the concept of Sustainability as a Service (MaaS) for reader who are not familiar with the concept".
R: The authors thank the reviewer for the appreciation. In section 3.4, a paragraph has been inserted which clarifies the meaning of MaaS for readers unfamiliar with this concept.
#2.7. "I think the paper will benefit greatly from a comprehensive review of grammar and spelling mistakes"
R: The new version of the paper has been sent to the publisher´s language editing services.
Round 2
Reviewer 2 Report
I am satisfied with the improvements made. Again, I would like to congratulate the authors for their work.